# Training Benefits and Injury Risks of Standing Yoga Applied in Musculoskeletal Problems: Lower Limb Biomechanical Analysis

**DOI:** 10.3390/ijerph18168402

**Published:** 2021-08-09

**Authors:** Ai-Min Liu, I-Hua Chu, Hwai-Ting Lin, Jing-Min Liang, Hsiu-Tao Hsu, Wen-Lan Wu

**Affiliations:** 1Department of Sports Medicine, College of Medicine, Kaohsiung Medical University, Kaohsiung 80708, Taiwan; amiliu1117@gmail.com (A.-M.L.); ihchu@kmu.edu.tw (I.-H.C.); whiting@kmu.edu.tw (H.-T.L.); taiga1115@gmail.com (J.-M.L.); 2Center for Physical and Health Education, Si-Wan College, National Sun Yat-Sen University, Kaohsiung 80424, Taiwan; hsuma@mail.nsysu.edu.tw

**Keywords:** yoga asana, biomechanics, electromyography, joint moments

## Abstract

Standing yoga poses strengthen a person’s legs and helps to achieve the goal of musculoskeletal rehabilitation, but inadequate exercise planning can cause injuries. This study investigated changes in the electromyogram and joint moments of force (JMOFs) of lower extremities during common standing yoga poses in order to explore the feasibility and possible injury risk in dealing with musculoskeletal problems. Eleven yoga instructors were recruited to execute five yoga poses (Chair, Tree, Warrior 1, 2, and 3). The results revealed significant differences in hip, knee, and ankle JMOFs and varying degrees of muscle activation among the poses. Among these poses, rectus femoris muscle activation during the Chair pose was the highest, Warrior 2 produced the highest muscle activation in the vastus lateralis of the front limb, while Warrior 1 had the highest muscle activation in the vastus medialis of the back limb. Therefore, all three poses can possibly be suggested as a therapeutic intervention for quadriceps strengthening. Warrior 1 was possibly suggested as a therapeutic intervention in order to reduce excessive lateral overload of the patella, but the possible adverse effects of Warrior 2 with the highest knee adductor JMOF in the back limb could raise joint reaction forces across the medial condyles. In single-leg balance postures, Warrior 3 had unique training effects on the hamstring, and is therefore suggested as a part of hamstring rehabilitation exercises. The Tree pose induced low lower-extremity JMOFs and a low level of thigh muscle activations when it was performed by senior instructors with excellent balance control; however, for yoga beginners with insufficient stability, it will be a useful training mode for strengthening the muscles that help to keep one upright. This study quantified the physical demands of yoga poses using biomechanical data and elucidated the structures and principles underlying each yoga movement. This is crucial for yoga practitioners.

## 1. Introduction

As a form of exercise, the practice of yoga has increased rapidly in recent years among all age groups, as individuals seek ways to stay healthy and resolve physiological and psychological problems. The benefits of yoga practice have been demonstrated in previous studies: yoga improves flexibility and strength [1,2,3,4], raises the quality of life, reduces the risk of cardiovascular disease [5], and improves musculoskeletal health [6,7,8,9,10]. It has been reported that yoga is safe as a usual form of care and exercise [11]. However, there are risks associated with practicing yoga incorrectly, including muscle strain, torn ligaments, and more severe injuries [12]. Forty percent of yoga-related injuries occur in the lower extremity, specifically the knee (41%), and 43% of injuries are sustained while practicing yoga [13]. Another study also reported that 74.2% of people had mechanical myofascial pain due to overuse, most commonly resulting in injuries that involved the hyperflexion and hyperextension of the spine [12]. Therefore, physicians or individuals should consider those risks when choosing yoga as their exercise.

Previous research has applied joint moments of force (JMOFs) in order to assess the physical demands of yoga. The calculation of moments requires the measurement of forces and moment arms about the joints. An evaluation of the joint moments generated during yoga allows the strengthening component of this exercise to be measured; however, excessive joint moments, or moments about joints where either the joint or the supporting tissues (muscle, tendon, ligament) are not evolved for and/or not conditioned to deal with them, have been suggested as being a source of injury. Therefore, this parameter connects abnormal movement to underlying muscle malfunction and bone misalignment. Previous studies have assessed the training load of yoga postures included in lower extremity strengthening programs and verified that most joint moments of the lower extremity were notably smaller in activities of daily living and notably larger in yoga [14]. Many joint moments generated during yoga are almost equivalent to those experienced during running [15]. Studies on the senior population [16,17,18] have also quantified the lower extremity JMOFs related to standing yoga postures in senior yoga programs in order to explore the long-held conception regarding the need for modifications of yoga poses for older adults. These studies demonstrated that the demands associated with some postures and posture modifications are not always intuitive.

For help in a discussion on patients regarding their rehabilitation program after ankle injury or surgery, Mears et al. (2019) used motion capture and force plates to assess the range of motion and joint moments of the ankle while practicing seven yoga poses that were considered to challenge the ankle joint [19]. Their findings in joint moments showed that the joint loading was highest during single leg poses. For OA knee populations, one study analyzed the biomechanical demands of standing yoga postures in patients with knee osteoarthritis (OA) and found that the knee adduction moments during yoga postures were lower than those during normal gait [20]. This implies that yoga is safe for patients with knee OA. Furthermore, Longpré et al. (2015) compared knee adduction moments across different static standing yoga postures in order to identify appropriate exercises for knee OA [21]. They found that squatting and lunging postures improved leg strength and potentially minimized exposure to high knee adduction moments.

Dynamic EMG results can also be used to assess whether strengthening exercises are effective. By examining EMG signals during yoga training, Salem et al. (2013) quantified the muscle activities of 21 Hatha yoga postures commonly used in senior yoga programs. They demonstrated that all postures elicited appreciable rectus abdominis activity, which was up to 70% of that induced while walking [16]. Ni et al. (2014) examined muscle activation patterns in selected trunk and hip muscles when performing 11 yoga poses. Their results indicated that the core muscle activation patterns depend on the trunk and pelvic positions during these poses [22]. Ni et al. (2014) also examined the activities of 14 dominant side trunk and upper and lower extremity muscles in 36 yoga practitioners during different yoga asanas across three skill levels. The results were presented as the root mean square of the EMG signal and indicated that different poses produced specific muscle activation patterns that varied depending on skill level [23]. Moreover, Kelley et al. (2018) examined the activities of five lower limb muscles in 13 experienced yoga practitioners during single-limb (Tree and Warrior 3) and double-limb (Downward Facing Dog, Half-Moon, and Chair) yoga asanas [24]. The EMG results showed differences in frontal and sagittal plane muscle activation between the single-limb and double-limb poses. In another study conducted 1 year later, they examined the muscle activation during single-limb yoga poses (Tree, Half-Moon, and Warrior 3) in comparison with a resting pose (mountain pose) [25]. They concluded that single-limb yoga poses require the increased use of the ankle musculature compared with the thigh musculature. It can be speculated that one-leg and two-leg yoga postures have different levels of lower limb muscle activation requirements.

Yoga is becoming popular as a supplement or alternative treatment for musculoskeletal diseases. However, rehabilitation training must be designed with disease treatment as the center concept, which means that the interventions and approach selected for each disease should depend on their goals and precautions. Although some biomechanical studies have assessed the training load of some yoga postures and compared it to the load of activities of daily living in order to let people understand the health benefits of yoga training, limited research has used biomechanical profiles, both JMOF and EMG, to assess yoga asanas and to give a practical suggestion for the treatment course. The present study aimed to determine the hip, knee, and ankle JMOFs across the sagittal and frontal planes and to differentiate the contributions between different muscles through the EMG output when experienced yoga instructors perform standard poses, which can provide guidelines for both yoga teachers in designing strength training courses for the general population or medical practitioners in helping patients returning from pathology or injury of the lower limb.

## 2. Materials and Methods

### 2.1. Participants

Eleven female yoga instructors were recruited from a yoga studio to voluntarily participate in this study. Their mean height was 159.2 cm (±3.0 cm), their mean weight was 51.0 kg (±4.6 kg), and their average age was 40.7 years (±6.0 years). Ten yoga instructors had Registered Yoga Teacher 200-h (RYT200) qualifications, and one had a Registered Yoga Teacher 500-h (RYT500) qualification. All instructors had at least 1000 h of yoga teaching experience. The average yoga teaching experience was 7.0 years (±3.7 years), and the average duration of continuous yoga practice was 8.9 years (±3.0 years). The instructors participated in self-practice for 7.3 h per week (±2.5 h) and led yoga classes 8.6 h per week (±4.0 h). The Institutional Review Board of Kaohsiung Medical University approved this experiment (KMUHIRB-E(II)-20170015). All participants were given an overview of this study and signed the informed consent form.

### 2.2. Equipment

Biomechanical analysis was performed at the motion analysis laboratory of Kaohsiung Medical University. Kinematic data were collected using a six-camera motion capture system (Qualysis ProReflex camera, Qualisys Track Manager with Oqus-CMOS, Qualysis AB, Sweden). The motion capture system was synchronized with a ground reaction force platform (Type 9286A, 9286AA, Kistler Instrument, Inc., Winterthur, Switzerland) to record data at a sampling frequency of 100 Hz for the motion capture system and 400 Hz for the force platform. Visual 3D software (Version 3.79, C-Motion, Inc., Germantown, MD, USA) was used to calculate external joint moments. Joint moments were calculated using an inverse dynamics approach and are expressed as joint moments normalized by body mass (Nm/kg). Electromyography ((Noraxon U.S.A. Inc., Scottsdale, AZ, USA) was used to detect muscle activity signals. Bipolar signals were recorded by pairs of 10 mm diameter Ag/AgCl surface disc electrodes placed at 20 mm (center-to-center). EMG signals were recorded at a sampling frequency of 1000 Hz, and data were collected simultaneously by the motion capture and force platform systems.

### 2.3. Yoga Asanas

Five common standing yoga asanas were tested in this study: Chair (Sanskrit name: Utkatasana), Tree (Vrksasana), Warrior 1 (Virabhadrasana 1), Warrior 2 (Virabhadrasana 2), and Warrior 3 (Virabhadrasana 3). The five standing yoga poses are illustrated in Figure 1, and the instructions of five asanas were as follows. Chair pose (Utkatasana): Stand with your feet approximately hip-width apart. When practicing it, one has to make sure the knees are pointing straight ahead and to press the shoulders down and back and stare at a point in front of you for balance. Tree pose (Vrksasana): Bend the rising leg knee and shift all of the weight into the supporting leg. The supporting leg must remain steadily on the floor. Warrior 1 pose (Virabhadrasana I): With the front limb knee directly over the ankle, straighten the back limb by pressing the heel towards the floor. The arms are over the head in an H position with the palms facing each other. Warrior 2 pose (Virabhadrasana II): Keep the front limb toes to the right direction and your front heel aligned with your back foot arch. Bend the knee directly over the ankle. Sink the hips down towards the floor, and keep the head up to lengthen the spine. Warrior 3 pose (Virabhadrasana III): Shift all of the weight onto the supporting leg and keep it straight. Raise the back leg up and keep the hips, trunk, arms, and head in one straight line for the final position.

### 2.4. Experimental Procedure

The electrode sites on each participant’s thighs were cleaned with 70% alcohol. Disposable bipolar electrodes were then placed over the muscle belly of the vastus lateralis (VL), rectus femoris (RF), vastus medialis (VM), bicep femoris (BF), and semitendinosus (SEMI) for collecting surface EMG signals. The participants were explained the manual muscle testing protocol for achieving their maximum voluntary contraction (MVC) for each of the five muscles tested. For manual muscle testing of MVC for the VL, RF, and VM, the participants straightened their legs against resistance in the supine position. MVC signals for the BF and SEMI were collected in the prone knee bend, in which the participants were pulled against resistance.

In order to collect motion capture data, reflective markers were placed on the following anatomical landmarks: the first and fifth metatarsal heads; calcaneus, medial, and lateral malleolus; femoral epicondyles; anterior superior iliac spine; and sacrum. Based on these markers, five standing yoga asanas were modeled, focusing on the following lower extremity segments: pelvic, thighs, shanks, and feet.

Subjects performed the yoga asanas on the force platform. The participant began in a starting position and moved into the posture. Once the participant moved into the position, they provided a verbal cue to the research associate to collect data. Each posture was held statically for 10 s and repeated three times. The posture order was randomized. Warrior 1 and Warrior 2 were performed with the right limb as the leading leg (front limb) and the left limb as the trailing leg (back limb). However, due to the asymmetric posture of these two asanas, the front limb data (right foot on the force platform) were collected first, followed by the collection of the back limb data (left foot on the force platform). The Chair, Tree, and Warrior 3 poses were performed only with the right limb placed on the force platform.

### 2.5. Data Analysis

Analyses of JMOFs for Warrior 1 and Warrior 2 were conducted for the left and right limbs separately. For Chair, Tree, and Warrior 3, JMOFs were only analyzed for the right limb. Therefore, data were collected for five leading limbs and two trailing limbs. Data collected during the middle 6 s of the static portion of each yoga asana were used for analysis. Hip, knee, and ankle JMOFs were normalized to each participant’s body mass. The right limb was used for all measures, along with the left limb for the Warrior 1 and Warrior 2 asanas. EMG data were normalized to each participant’s peak activity during MVC by using MyoResearch XP Basic Edition 1.07.01 software (Noraxon U.S.A. Inc., Scottsdale, AZ, USA), and bandpass filtering between 80 and 250 Hz was applied.

### 2.6. Statistical Analysis

The data were analyzed using one-way repeated measure ANOVA. The independent variables were the five yoga asanas (front and back limbs were considered separately for Warrior 1 and Warrior 2). The ankle, knee, and hip JMOFs and EMG values of the standing limb were the dependent variables. Post hoc tests were used when significant main effects were found. Statistical analyses were completed using SPSS (Version 20, IBM, Armonk, NY, USA). The statistical significance level was set at 0.05.

## 3. Results

The mean JMOFs in the sagittal and frontal planes for the hip, knee, and ankle are presented in Table 1. The hip JMOFs in the sagittal and frontal planes, knee JMOFs in the sagittal and frontal planes, and ankle JMOFs in the sagittal plane were significantly different (*p* < 0.05) across the five yoga asanas. The JMOFs of the front and back limbs were analyzed separately for Warrior 1 and Warrior 2. Average EMG activities are presented in Table 2 and are expressed as a percentage of activity achieved during MVC (% MVC) for each yoga asana. No significant differences were found in EMG values obtained from these yoga asanas (*p* > 0.05). We, therefore, used the maximum value of descriptive statistics for data interpretation.

### 3.1. Hip JMOFs in the Sagittal and Frontal Planes

In the sagittal plane (Figure 2A), significant differences were found in the hip flexor/extensor JMOFs of the seven limbs in post hoc tests, except for between the front limb for Warrior 2 and Warrior 3. In the frontal plane (Figure 2B), the post hoc results showed significant differences in the hip adductor/abductor JMOFs of the seven limbs, except for between the back limb for Warrior 1 and Tree.

### 3.2. Knee JMOFs in the Sagittal and Frontal Planes

Significant differences were observed in the knee extensor JMOFs in the Chair pose (*p* < 0.01), and the JMOF was approximately four times greater in the Chair pose than in the front limbs for Warrior 1 and Warrior 2 (Figure 2C). The knee flexor JMOFs, in order from the highest to lowest, were as follows: Warrior 1 back limb, Warrior 2 back limb, Tree, and Warrior 3. The knee flexor JMOF of the back limb for Warrior 1 was approximately four times greater than that of the other three poses, and this difference was statistically significant (*p* < 0.01). However, post hoc tests revealed that the Warrior 2 back limb JMOF was not significantly different from the Tree and Warrior 3 JMOFs. Moreover, no significant difference was found between the Tree and Warrior 3 JMOFs.

The back limb for Warrior 2 generated the highest knee adductor JMOF (Figure 2D), which was significantly different from that of the other six limbs (*p* < 0.01). No significant differences were found in the JMOFs of Tree, back limb for Warrior 1, and Warrior 3. The JMOF of Warrior 3 was significantly different from that of the front limb for Warrior 1, front limb for Warrior 2, and Chair. However, no other significant differences were found in the JMOFs of the front limb for Warrior 1, front limb for Warrior 2, and Chair in post hoc tests.

### 3.3. Ankle JMOFs in the Sagittal and Frontal Planes

All analyzed yoga asanas produced ankle plantarflexion JMOFs (Figure 2E). The Warrior 2 front limb ankle plantarflexion JMOF was significantly higher than Warrior 1 back limb and Warrior 2 back limb JMOFs (*p* < 0.01). In the frontal plane, only the back limb for Warrior 2 produced an ankle evertor (+Abd) JMOF; ankle invertor (-Add) JMOFs were generated for the other six standing limbs for each yoga asana (Figure 2F). However, the differences between ankle JMOFs in the frontal plane of these yoga asanas did not reach significance in Bonferroni tests.

### 3.4. Average EMG Activity (%MVC)

Table 2 shows that VL muscle EMG activity was the highest in Warrior 2 front limb EMG activity (47.7% MVC). VL muscle activity was the second highest in the Chair pose (47.2% MVC) and was similar to Warrior 2 front limb EMG activity. However, in the RF, EMG activity was highest in the Chair pose (33.0% MVC). RF muscle activity was the second highest in the back limb for Warrior 1 (32.5% MVC), and VM EMG activity was the highest in the back limb for Warrior 1 (50.4% MVC). EMG activities in the BF (28.5% MVC) and SEMI (37.2% MVC) muscles were the highest during Warrior 3.

## 4. Discussion

This study identified the JMOFs and average EMG activity of the lower extremities during five common standing yoga asanas (including the back limbs for Warrior 1 and Warrior 2) in a sample of yoga instructors.

### 4.1. Squat Pose: Chair

In this study, the Chair asana produced the higher knee extensor JMOF (0.041 Nm/kg) and lowest knee adductor JMOF (0.001 Nm/kg). Quadriceps EMG activities were also higher during the Chair asana (33.0–47.2% MVC). This finding indicates that the Chair asana should be included in knee strengthening exercise programs. Compared with other standing yoga, it is a more suitable training choice for those with symptomatic knee OA [20], as it is effective for preventing excessive medial knee joint loading and minimizing the knee adductor JMOF, as well as maximizing the training effect on the muscle. The Chair pose may also be beneficial for general exercise. This suggestion was made in previous studies for healthy [21,24] and senior populations [16,17], which found that squat exercises tone the leg muscles excellently. Moreover, this asana was suggested to stretch the calf muscles and to lift the inner arch for reducing symptoms of flat feet.

### 4.2. Lunge Pose: Warrior 1 and Warrior 2

The specific biomechanics of the two high lunges varied between Warrior 1 and Warrior 2. Both postures are fundamental standing yoga poses. The greatest hip extensor JMOF (0.288 Nm/kg) was generated for the front limb for Warrior 1, and the greatest hip flexor JMOF (0.316 Nm/kg) was generated for the back limb. For Warrior 2, the front limb generated the highest hip adductor JMOF (0.323 Nm/kg), and the back limb produced the highest hip abductor JMOF (0.336 Nm/kg). This result is not surprising; greater joint moment development is associated with a longer moment arm; the widened stance of the Warrior 1 and Warrior 2 postures lengthens the moment arm of the force, with respect to the hip joint center in the sagittal and frontal plane individually.

A greater JMOF may put a greater load on the hip joint. One study has evaluated muscular activation while performing standing yoga poses for those with hip impingement, and the results showed that patients with hip pain [26] undergo more rapid periarticular muscular fatigue than control subjects when performing weight-bearing yoga poses, which may reduce their protecting effect on the hip joint. Another study also showed that the daily cumulative hip moment is associated with the radiographic progression of secondary hip OA [27]. Therefore, people with hip pain, a hip impingement symptom, or those diagnosed with hip OA should avoid the lunge pose, such as Warrior 1 and Warrior 2, especially for those who have pain while performing these poses, which can worsen the disease. In addition, osteoporosis is one of the most common diseases of the musculoskeletal system in old age. General guidelines for yoga practice in older adults with hip osteoporosis are that it must include resistance and balance training; however, the contraindicated movements include the end-range internal/external rotation of the hip. Therefore, before practicing yoga, such as Warrior 2, one should be appropriately counselled regarding the potential risk of symptomatic exacerbation and possible counterproductive effects of participation.

For the knee joint, the back limb for Warrior 1 produced a higher knee flexor JMOF (0.075 Nm/kg) than the back limb for Warrior 2 (0.026 Nm/kg) in the sagittal plane. Furthermore, the back limb for Warrior 2 generated a significantly higher knee adductor JMOF (0.065 Nm/kg) than the Warrior 1 back limb knee adductor JMOF (0.022 Nm/kg) in the frontal plane. These results are similar to those from a previous study [17], which observed that the knee adductor JMOF of the Warrior 2 back limb was greater than the JMOFs produced by other poses, and was 267% greater than the peak JMOF produced during self-selected walking. Therefore, the back limb in the Warrior 2 pose, which indicated higher knee adductor torque, may impact the medial side of the knee joint. This information is crucial when practicing yoga asanas.

In this study, the quadriceps muscles (RF, VM, and VL) had high EMG activity during the Warrior 1 and Warrior 2 poses. VM muscle activation was the highest during the Warrior 1 back limb, and VL muscle activation was the highest during the Warrior 2 front limb. The function of VM is to realign the patella during a knee extension. A muscular balance between VL and VM is important. Where VM is weaker, the patella is pulled too far laterally, which can cause increased contact with the condylus lateralis, leading to degenerative changes underneath the kneecap. The restoration of adequate VM strength and function is an essential factor for achieving good recovery. Therefore, Warrior 1 may be recommended as a therapeutic intervention for patients, such as patellofemoral pain syndrome or chondromalacia patellae, by using a pressing down workout of the rear leg in order to increase the muscle strength of VM.

### 4.3. Single-Leg Balance Poses: Tree and Warrior 3

In single-leg balance postures, such as the Tree pose and Warrior 3 pose, the ground reaction force passes almost directly through the center of the hip joint. Accordingly, we observed low hip JMOFs in the sagittal plane and frontal plane in these two postures. However, these findings do not necessarily apply to the general population. Senior instructors have good balance control in these single-leg balance postures; thus, the joint moment is low in each joint or plane. Senior yoga practitioners with unstable standing balance may benefit from practicing these poses near a wall or chair in order to provide lateral stability. Previous research also showed that ankle joint loading was highest in single-leg poses [19]. Patients with a history of ankle injury, along with the degeneration of cartilage, should be cautious when performing these poses. On the contrary, there are many physical benefits to these single-leg balance poses; for example, the Tree pose is the first balance pose that most yoga beginners learn, and Warrior 3 could strengthen the whole back side of the body, including the shoulders, hamstrings, calves, ankles, and back. It also tones and strengthens the abdominal muscles. This study has verified that Warrior 3 has unique training effects on the hamstring. Weak hamstrings are more likely to cause musculoskeletal injuries, such as increasing the risk of a recurrent hamstring strain injury or increasing pain or tightness in the lower back. Therefore, integrating this yoga practice into a rehabilitation program is helpful for such patients.

### 4.4. Integrate Gender and Age Concerns in Yoga Practice

A limitation of the study was that it did not include male participants. The body types of men and women are different; for example, the acetabulum faces more laterally in males, whereas it faces more anteriorly in females. A previous study [28] has verified that this affects the hip joint moments of healthy older walkers. Greater external hip adduction and internal rotation, along with hip extension moments, were found for females compared to males. As external joint moments are surrogate measures of the joint contact forces, the results of this study suggest that the hip joint stress for the female population is higher compared to the male population. This could contribute to a greater joint degeneration at the hip in females as compared with males. If we truly want to be anatomically correct, then it becomes obvious that women need to perform their standing yoga poses differently to men. In poses such as Warriors 1 and 2, instead of aligning the feet as one is typically instructed to (with your front heel aligned with your back arch), try widening the stance a bit by aligning the front heel with the back heel; otherwise, if the base of support is too narrow side-to-side, then many accommodations will occur with the torquing and twisting of various body parts that could lead to injury over time.

Although this study did not include elderly subjects, previous research [19] indicated that the younger population experienced larger joint moments about both ankles while performing common standing yoga poses than their older counterparts, perhaps suggesting different load distribution strategies by the two populations. EMG activation and the coordination of ankle muscles during balance tasks also change with age [29]. In order to compensate for the deterioration of the postural control system and provide them with a posture that has greater stability, elders showed increased co-activation of the agonist and antagonistic muscles in the ankle joint when the support plane was changed. However, this strategy could cause greater body asymmetry, which consequently leads to balance disorders and falls. Since the age and fitness level might arise under different joint movement or muscle activation strategy, recommendations for integrating yoga into a health and wellness program must depend on participants’ individual fitness levels and must follow the sequence selection from static to dynamic forms of balance yoga practicing.

## 5. Conclusions

This study quantified the lower extremity physical demands of five common standing yoga asanas and used them to give suggestions for improving the strength training in various musculoskeletal conditions. Our findings suggest that the Chair pose was the better choice for knee OA patients, because it strengthens the quadriceps and minimizes the knee adduction moment. Warrior 1 is recommended for chondromalacia patellae patients, due to its stronger strengthening effects on VM. Patients with osteoporosis of the hip may have to avoid Warrior 2, because of extreme hip external rotation. For hip OA patients, neither Warrior 1 nor 2 is suitable, because of their huge hip JMOFs. Our results also suggest that Warrior 3 has positive effects on hamstring training, and it should be the first choice of these five asanas for the rehabilitation of recurrent hamstring strain patients. We recommend that senior yoga practitioners with unstable standing balance should practice the Tree pose next to a wall or chair in order to provide lateral stability. For those with problems of degeneration of cartilage in the ankle joint, the risk of placing stress on one foot should be considered when selecting yoga exercises. In addition to the above cases, it is indeed a good choice for beginners’ standing balance training.

## Figures and Tables

**Figure 1 ijerph-18-08402-f001:**
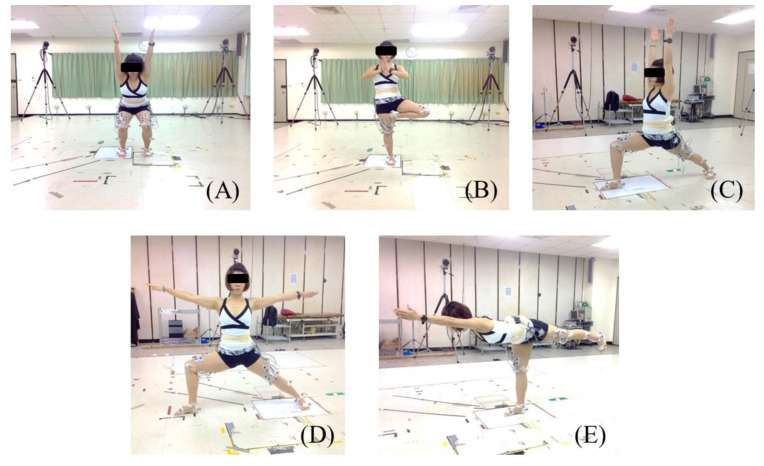
The five standing yoga poses: (**A**) Chair pose (Utkatasana); (**B**) Tree pose (Vrksasana); (**C**) Warrior 1 pose (Virabhadrasana I); (**D**) Warrior 2 pose (Virabhadrasana II); (**E**) Warrior 3 pose (Virabhadrasana III).

**Figure 2 ijerph-18-08402-f002:**
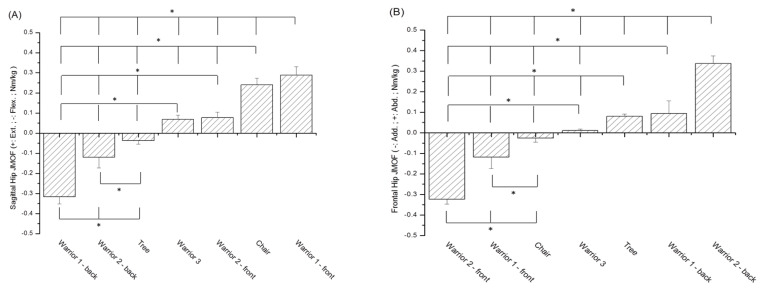
(**A**) Comparison of the hip JMOFs of the seven limbs in the sagittal plane. * denotes a significant difference in Bonferroni post hoc tests; (**B**) comparison of the hip JMOFs of the seven limbs in the frontal plane. * denotes a significant difference in Bonferroni post hoc tests; (**C**) comparison of the knee JMOFs of the seven limbs in the sagittal plane. * denotes a significant difference in Bonferroni post hoc tests; (**D**) comparison of the knee JMOFs of the seven limbs in the frontal plane. * denotes a significant difference in Bonferroni post hoc tests; (**E**) comparison of the ankle JMOFs of the seven limbs in the sagittal plane. * denotes a significant difference in Bonferroni post hoc tests; (**F**) comparison of the ankle JMOFs of the seven limbs in the frontal plane. * denotes a significant difference in Bonferroni post hoc tests.

**Table 1 ijerph-18-08402-t001:** Basic descriptive statistics for the JMOFs in sagittal and frontal planes for the hip, knee, and ankle. Data are presented as mean (standard deviation).

		Chair	Tree	Warrior 1		Warrior 2		Warrior 3
				Front limb	Back limb	Front limb	Back limb	
**Hip**	Sagittal plane * (+:Ext.; −:Flex.)	0.240 (0.034)	−0.037 (0.017)	0.288 (0.042)	−0.316 (0.036)	0.077 (0.028)	−0.120 (0.053)	0.069 (0.021)
	Frontal plane * (−:Add.; +:Abd.)	−0.026 (0.021)	0.080 (0.011)	−0.118 (0.056)	0.094 (0.061)	−0.323 (0.023)	0.336 (0.037)	0.012 (0.007)
**Knee**	Sagittal plane * (+:Flex.; −:Ext.)	−0.041 (0.012)	0.015 (0.012)	−0.013 (0.013)	0.075 (0.023)	−0.009 (0.011)	0.026 (0.018)	0.013 (0.010)
	Frontal plane * (+:Add.; −:Abd.)	−0.001 (0.015)	0.022 (0.007)	0.001 (0.011)	0.022 (0.059)	0.000 (0.010)	0.065 (0.078)	0.018 (0.005)
**Ankle**	Sagittal plane * ( +:Plantar flexor; −:Dorsiflexion)	0.008 (0.003)	0.008 (0.001)	0.008 (0.003)	0.006 (0.002)	0.008 (0.001)	0.006 (0.001)	0.008 (0.001)
	Frontal plane (−: Invertor.; +: Evertor)	−0.001 (0.001)	−0.001 (0.001)	−0.001 (0.002)	−0.001 (0.001)	−0.001 (0.001)	0.001 (0.001)	−0.001 (0.001)

* denotes a significant difference between the five yoga asanas, *p* < 0.05.

**Table 2 ijerph-18-08402-t002:** Average EMG activity (% MVC).

		Vastus Lateralis	Rectus Femoris	Vastus Medialis	Biceps Femoris	Semitendinosus
**Chair**		47.2 ± 24.0	**33.0 ± 10.1**	35.0 ± 11.2	9.5 ± 5.2	10.6 ± 7.8
**Warrior 1**	**front limb**	38.9 ± 20.7	24.5 ± 15.6	27.7 ± 4.8	10.7 ± 6.5	13.8 ± 9.9
	**back limb**	14.6 ± 6.6	32.5 ± 11.0	**50.4 ± 11.7**	13.8 ± 7.8	19.1 ± 14.7
**Warrior 2**	**front limb**	**47.7 ± 26.0**	19.7 ± 7.4	32.0 ± 2.4	12.5 ± 8.8	10.7 ± 6.4
	**back limb**	8.5 ± 2.8	21.7 ± 9.6	32.9 ± 10.7	13.0 ± 8.8	30.6 ± 14.3
**Warrior 3**		25.9 ± 11.1	14.0 ± 5.4	18.0 ± 5.6	**28.5 ± 10.0**	**37.2 ± 21.1**
**Tree**		31.4 ± 19.2	25.7 ± 11.3	27.9 ± 21.5	11.0 ± 5.4	11.1 ± 10.7

Note: Peak EMG activity observed in each muscle for the five yoga asanas (seven limbs) is written in bold.

## Data Availability

The data presented in this study are available on request from the corresponding author.

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
