# Peer review of "Training Benefits and Injury Risks of Standing Yoga Applied in Musculoskeletal Problems: Lower Limb Biomechanical Analysis"

_ijerph, 2021, doi:10.3390/ijerph18168402_

Round 1
Reviewer 1 Report
Thank you for the opportunity to review this manuscript. The present study aimed to determine the hip, knee, and ankle JMOFs across the sagittal and frontal planes and the EMG output that are generated by yoga instructors when they perform specific yoga asanas.
Please explain:
1) why MoCap data was not presented
2) why the authors did not take into raport the transverse plane
3) the literature should be expanded to include the study of yoga semg and muscle activity in people of different ages (e.g. Meng Ni, Kiersten Mooney, Kysha Harriell, Anoop Balachandran, Joseph Signorile, Core muscle function during specific yoga poses,
Complementary Therapies in Medicine, Volume 22, Issue 2,
2014,Pages 235-243, Błaszczyszyn, M.; Szczęsna, A.; Piechota, K. sEMG Activation of the Flexor Muscles in the Foot during Balance Tasks by Young and Older Women: A Pilot Study. Int. J. Environ. Res. Public Health 2019, 16, 4307).
Reviewer 2 Report
While I feel that this paper could be of interest, I think that it needs some work to stand up to the standards of the journal. At first sight (title and abstract) it seems to be a simple biomechanical analysis of standing yoga poses, so some parts of the paper are not quite clear with respect to this context. Please clarify the goal of the paper not only in the main body, but also in the title and in the abstract, and adapt the paper to the specific goal to increase clarity.
Introduction: There is great emphasis on studies on patients with osteo-arthritis, but it is not clear how this connects to the study of alignment in yoga instructors. While it is true that this is a group at greater risk for injuries, they are not the only individuals that would benefit from a correct alignment during asanas. Thus it is not clear why to concentrate this much on this specific group. I would suggest starting with a wider overview of previous studies and maybe delve into specific groups after that. Moreover, since it seems that the authors are quite interested in possible aspects that could increase the risk of injuries, I suggest to briefly address the possible issues specifically deriving from this kind of postures.
Yoga asanas: Please describe the poses. While there are illustrations, some important aspects of the poses may not be evident. Moreover, poses may have slight modifications based on yoga style or based on which poses they are connected to (one example is Chair pose, which is practiced with feet together in Ashtanga, but may be modified with feet hips-width apart to increase balance). Also feet alignment may be important here (i.e. heel to heel, heel to arch) since it may slightly modify the pose, especially for hip rotations in the warrior poses.
Experimental procedure: Why was the right limb selected as leading leg? While years of training should help balancing out the differences, it is a known fact that the human body has inherently a stronger and a weaker side, which also influences flexibility.
Discussion: Please add also a more general discussion especially regarding the important aspects to keep in mind not only for high-risk groups but also for the general public to minimize the risk of injury. At the moment it is more a description of the results than a proper discussion. Moreover, highlight the limitations of the work and how one could derive useful information from the analysis of the asanas performed by well-trained participants to help reducing injuries in other groups. Also speculate about the differences that could be present in male yogis compared to female, especially based on the anatomical differences in particular of the pelvis and in muscle strength and flexibility.
Round 2
Reviewer 2 Report
Dear authors,
Thank you for addressing all the raised points. I have no further comments about the contents.
Nevertheless, a spell-check is still needed (for example at p. 1 line 24 "[...] to reduce excessive lateral overload of the patellar, but [...]").
Please check the article for corrections.